# In Vitro and In Silico Evaluation of Antiproliferative Activity of New Isoxazolidine Derivatives Targeting EGFR: Design, Synthesis, Cell Cycle Analysis, and Apoptotic Inducers

**DOI:** 10.3390/ph16071025

**Published:** 2023-07-19

**Authors:** Fahad Alminderej, Siwar Ghannay, Mohamed Omer Elsamani, Fahad Alhawday, Abuzar E. A. E. Albadri, Serag Eldin I. Elbehairi, Mohammad Y. Alfaifi, Adel Kadri, Kaïss Aouadi

**Affiliations:** 1Department of Chemistry, College of Science, Qassim University, Buraidah 51452, Saudi Arabia; f.alminderej@qu.edu.sa (F.A.); s.ghannay@qu.edu.sa (S.G.); 431114194@qu.edu.sa (F.A.); aa.albadri@qu.edu.sa (A.E.A.E.A.); 2Department of Chemistry, Faculty of Science and Arts of Baljurashi, Al-Baha University, P.O. Box 1988, Albaha 65527, Saudi Arabia; mhmdomer6@gmail.com (M.O.E.); lukadel@yahoo.fr (A.K.); 3Department of Food Science and Technology, Faculty of Sciences, Omdurman Islamic University, Omdurman P.O. Box 382, Sudan; 4Department of Biology, Faculty of Science, King Khalid University, Abha 9004, Saudi Arabia; serag@kku.edu.sa (S.E.I.E.); alfaifi@kku.edu.sa (M.Y.A.); 5Cell Culture Laboratory, Egyptian Organization for Biological Products and Vaccines, VACSERA Holding Company, Giza 2311, Egypt; 6Department of Chemistry, Faculty of Science of Sfax, University of Sfax, B.P. 1171, Sfax 3000, Tunisia; 7Department of Chemistry, Laboratory of Heterocyclic Chemistry Natural Product and Reactivity/CHPNR, Faculty of Science of Monastir, University of Monastir, Avenue of the Environment, Monastir 5019, Tunisia

**Keywords:** molecular dynamics, molecular docking simulation, isoxazolidine, anticancer, EGFR, apoptosis, cell cycle analysis, ADME

## Abstract

A series of novel enantiopure isoxazolidine derivatives were synthesized and evaluated for their anticancer activities against three human cancer cell lines such as human breast carcinoma (MCF-7), human lung adenocarcinoma (A-549), and human ovarian carcinoma (SKOV3) by employing MTT assay. The synthesized compounds were characterized by NMR and elemental analysis. Results revealed that all the synthesized compounds displayed significant inhibition towards the tested cell lines. Among them, **2g** and **2f**, which differ only by the presence of an ester group at the C-3 position and small EDG (methyl) at the C-5 position of the phenyl ring (**2g**), were the most active derivatives in attenuating the growth of the three cells in a dose-dependent manner. The IC_50_ for **2g** were 17.7 ± 1 µM (MCF-7), 12.1 ± 1.1 µM (A-549), and 13.9 ± 0.7 µM (SKOV3), and for **2f** were 9.7 ± 1.3µM (MCF-7), 9.7 ± 0.7µM (A-549), and 6.5 ± 0.9µM (SKOV3), respectively, which were comparable to the standard drug, doxorubicin. The enzymatic inhibition of **2f** and **2g** against EGFR afforded good inhibitory activity with IC_50_ of 0.298 ± 0.007 μM and 0.484 ± 0.01 µM, respectively, close to the positive control, Afatinib. Compound **2f** arrested the cell cycle in the S phase in MCF-7 and SKOV3 cells, and in the G2/M phase in the A549 cell; however, **2g** induced G0/G1 phase cell cycle arrest, and inhibited the progression of the three cancer cells, together with significant apoptotic effects. The docking study of compounds **2f** and **2g** into EGFR ATP-active site revealed that it fits nicely with good binding affinity. The pharmacokinetic and drug-likeness scores revealed notable lead-like properties. At 100 ns, the dynamic simulation investigation revealed high conformational stability in the EGFR binding cavity.

## 1. Introduction

Cancer is a large group of chronic, non- communicable diseases (NCDs) that are characterized by uncontrolled cell proliferation and overexcited cell differentiation and division. It is expected to be the second leading cause of death after cardiovascular disease, with currently approximately 8 million deaths per year, and projected to reach 15 million by 2030, according to estimations [1,2]. Among them, lung cancer remains one of the greatest public health challenges in the world causing panic and confusion, and still appears as a real disaster for health systems and humanity worldwide, with approximately 27.5 million new cases estimated each year by 2040, followed by breast cancer as the second leading cause of death in women, affecting 2.1 million per year. In addition, the incidences of breast and ovarian cancers were estimated to make up 15% and 4%, respectively, of all deaths associated with cancers in women. In Saudi Arabia, the crude incidence rates of all cancers were 56.6/100,000 in males and 74.5/100,000 in females, with breast cancer ranked as the most common malignant cancer accounting for 17.7% of all reported cancers and 30.9% of all cancers recorded among women of all ages [3,4,5].

Cancers have spread widely as a result of genetic changes that occurred in aberrant EGFR activity that has been reported to be involved in the development of many human cancers. Therefore, the epidermal growth factor receptor (EGFR) still remains a crucial targetable transmembrane protein in cancer management known for its role in cell proliferation, survival, migration, adhesion, and differentiation [6]. EGFR has been clinically proven as a valuable therapeutic target for non-small cell lung cancer (NSCLC) with gefitinib and erlotinib being the most approved first-generation EGFR tyrosine kinase inhibitors [7,8]. Current ineffective treatments against cancer including surgery, immunosuppression chemotherapy, and radiotherapy have many adverse effects and high drug resistance accompanied with high rates of incidence, mortality, and morbidity [8,9]. In contrast, chimeric antigen receptor T (CAR-T) cells and antibody–drug conjugates (ADCs) currently remain the most popular immunotherapies strategies in oncology for anticancer drug development [10,11]. Therefore, there is a need to discover and develop a new and effective anticancer agent with selectivity to kill this anarchic development of cells or inhibit their proliferation with fewer side effects.

In this context, heterocyclic compounds containing at least two different heteroatoms have aroused the interest of chemists, in particular because of their diverse potential therapeutic applications. Among them, nitrogen- and/or oxygen-containing heterocyclic compounds, especially isoxazolidine derivatives, have received special attention because of their promising results as antitumor and antiviral agents [12,13]. Also, the isoxazolidine cycle is known to be a good mimic of the furanose and/or ribose cycle to access modified nucleosides which have shown encouraging results with interesting antiviral and anticancer properties [14,15,16,17,18] (Figure 1).

Enthused by the aforementioned drawbacks and in continuation of our research efforts on designing potent bioactive molecules [19,20,21], we have designed and synthesized some new enantiopure isoxazolidine derivatives. All compounds were evaluated for their in vitro anticancer activities towards MCF-7, A549, and SKOV3. Additionally, the most potent analogues were selected for studying mechanistic pathways such as EGFR assay, cell cycle analysis, and apoptosis markers to explore if the cytotoxic ability is accompanied by modification in cell cycle analysis and apoptosis induction. Finally, MD and DS were performed on the most active compounds, to understand the expected mode of action of the target compounds with EGFR active sites, in parallel with their draggability.

## 2. Results and Discussion

### 2.1. Chemistry

As shown in Figure 1, alkenes **1a**–**1g**, commercially available substrates, react with a menthone-derived nitrone under microwave irradiation for 2 h, at 130 °C and a power of 280 W, to give enantiopure isoxazolidines **2a**–**2g** as a single regioisomer where the nitrone oxygen reacts with the most substituted carbon of the alkene in an *exo* approach.

Isoxazolidine derivatives **2a**–**g** were obtained with the simultaneous creation of two stereogenic centers. All of the compounds **2a**–**f** are new except **2g** [22]. The stereochemistry has been proved by the spectroscopic usual methods (1H, 13C NMR, and NOESY 2D analysis). In our old work, we have shown that the coupling constant for H3 and H4a protons in the anti-position is low (^3^J_3-4b_ (anti) ≤ 3.7 Hz). Whereas, the syn ^3^J_3-4b_ coupling constant is larger (^3^J_3-4a_ (syn) ≥ 7.0 Hz) [23,24]. For the protons in the anti-position such as H4b and H5, the coupling constant exceeds 7 Hz. The interpretation of the 1H NMR spectra of the synthesized compounds **2a**–**g** revealed the presence of a doublet at 3.95 to 4.03 ppm attributed to the proton H3 with a coupling constant (^3^J_3-4a_ (syn)) close to 8 Hz. The elucidation of stereochemistry was determined using the 2D NOESY experiment. Indeed, the interpretation of the NOESY spectrum of compound **2b** proved the presence of strong correlations between H3–H4b and H5–H4b protons, weak correlations between the H3–H4a protons, and medium correlations between the H3–H methyl and H3–H isopropyl protons. These data confirm the proposed stereochemistry in Figure 2.

### 2.2. Biological Evaluation

#### 2.2.1. In Vitro Antiproliferative Activity

Compounds **2a**–**g** were tested for their growth inhibitory effectiveness against MCF-7, A-549, and SKOV3 cell lines by means of MTT assay, with doxorubicin used as an anticancer standard drug. The obtained results (Table 1) expressed as IC_50_ values were in the range of 7.2 ± 0.2–66.8 ± 2.7 µM, 9.7 ± 0.7–53.5 ± 1.1 µM and 6.5 ± 0.9–27.9 ± 1.9 µM, respectively, for MCF-7, A-549, and SKOV3. The preliminary results indicated that most of the examined compounds are active in inhibiting the tested cancer cells with **2f** and **2g** being the most effective compounds displaying very strong anticancer activity when compared to the positive control. Remarkably, we note the high antiproliferative activity of **2c** and **2d**, only towards SKOV3. In addition, the other compounds were found to possess moderate potencies depending on the tested cell lines.

Based on the above data, to explore the SAR study, it is worth mentioning that the substituents on aromatic rings affect the electron density on the ring which possess a significant effect on binding affinity and biological activity. In parallel with the implication of isoxazolidine scaffold’s impressive potential as a mimic of nucleosides, carbohydrates, PNA (peptide nucleic acid), amino acids, and steroid analogs, it has naturally emerged as a potential candidate in the field of anticancer drug discovery [15]. Results revealed that **2f** bearing the 2-hydroxyl group attached to the phenyl ring displayed potent activity against the three cell lines. Incorporation of a small EDG (methyl) at position C-3 of the aryl moiety (**2c**) showed lesser activity than **2f** for MCF-7 and A549 cancer cell lines, but better activity towards SKOV-3. Introducing an acetyl group at the position C-6 (**2b**) resulted in reduced activity. Meanwhile, the presence of an ester group at the position C-3 and a small EDG (methyl) at position C-5 of the phenyl ring (**2g**) improves the inhibitory growth over the three cells, suggesting the high contribution of the methyl 2-hydroxy-5-methylbenzoate group on the activity. Comparing **2a**, **2d**, and **2e**, which differ only by the presence of the hydroxyl group instead of methoxy and acetate groups at C-4, respectively, they showed a decrease in the activity towards MCF-7 and A549 cancer cell lines, due to the weaker effect of the methoxy group than the hydroxyl group on the aromatic ring. This effect was more pronounced with the acetate group that greatly decreased the activity.

#### 2.2.2. Epidermal Growth Factor Receptor Activity (EGFR) Inhibition

Furthermore, compounds **2f** and **2g** which are potent antiproliferative agents with the strongest inhibitory activity against the tested three cell lines, were further selected for determining their in vitro inhibitory effect against EGFR tyrosine kinase. Results showed that **2f** and **2g** displayed high inhibitory activity, with **2f** inhibiting EGFR at IC_50_ value of 0.298 ± 0.007 μM, whereas **2g** inhibited EGFR at IC_50_ value of 0.484 ± 0.01 which was close to the positive standard, Afatinib (IC_50_ = 0.074 ± 0.001 µM).

#### 2.2.3. Cellular Mechanism of Action

As a result of the encouraging cytotoxic effect and the EGFR inhibitory activity displayed by the top potent compounds **2f** and **2g** their cellular mechanism of action via treatment on cell cycle arrest or apoptosis was examined.

##### Apoptosis Study

The most active compounds **2f** and **2g** were subjected to cellular apoptosis assay towards the tested three cancer cell lines at IC_50_ values for 48 h, with doxorubicin as a positive control (Figure 3). Exposure of MCF-7 cells to **2f** and **2g** induce cellular apoptosis of 86.89% and 75.17%, respectively, when compared to that of positive control, 92.06% with a significant increase in both cell percentages of early (70.48% and 46.49%) and late (15.78 and 20.51%) apoptosis phases. Subjecting the cancer cell line A549 to compounds **2f** and **2g** showed a potent cell death of 92.61% towards **2f** which was comparable to that of doxorubicin (98.84%), while **2g** induced cellular apoptosis of 87.76%. Simultaneously, when compared with untreated cells, in A549 cancer cells, compound **2f** was found to cause early apoptosis (16.72%) and enhance late apoptosis (70.50%), while compound **2g** induced early apoptosis (60.72%) and reduced late apoptosis (17.69%). Surprisingly, the treatment of SKOV-3 cells by **2g** revealed similar cellular apoptosis (96.65%) to the anticancer drug (96.75%). However, **2f** caused 91.22% of cellular apoptosis, with significant elevations in early apoptosis for both **2f** (17.90%) and **2g** (12.92%) and increased the late apoptosis (67.16%) and (81.09%), respectively. Therefore, from the above-mentioned results, we concluded that compound **2f** can be considered an apoptotic inducer, as well as **2g**, towards MCF7 and A549, with results that are highly comparable to the positive control. Meanwhile, against SKOV3 **2g** was found to induce better cellular apoptosis (Table 2).

##### Flow Cytometric Cell Cycle Analysis

Based on the antiproliferative activity data shown above, a flow cytometry analysis was performed to identify the cell cycle arrest induced by the selected compounds (**2f** and **2g**) in the three cell lines, for 48 h at their respective IC_50_ concentrations (Figure 4). The cell cycle (G1, S, and G2) was analyzed after staining the DNA with propidium iodide (PI). Figure 4 infers that the treatment of the MCF-7 cell line with compound **2f** showed that the distribution of cells decreased in the G0/G1 (19 ± 1.2% vs. 31.3 ± 1.4%) and G2/M (17.7 ± 0.8% vs. 24.9 ± 0.7%) phases, and increased in the S phase (63.3 ± 0.6% vs. 43.7 ± 1.5%) when compared to the control. Upon exposure, the MCF-7 cell line treated with compound **2g** showed that the population of cells increased from 31.3 ± 1.4% to 53.7 ± 0.6% in the G0/G1 phase and decreased from 43.7 ± 1.5% to 36.3 ± 0.6% and from 24.9 ± 0.7% to 10 ± 0.1% in both the S and G2/M phases, respectively. Comparing to the control and after treatment of A549 separately with compounds **2f** and **2g**, the results showed that **2f** exerted a decrease in the percentage of cells at the G0/G1phase from 31.9 ± 0.2% to 23.1 ± 0.8% and an increase in the population at the G2/M phase from 24.7 ± 0.3% to 32.5 ± 0.4%, however, at the S phase, the distribution of cells was slightly affected. Additionally, A549 cells revealed that **2g** increased the population from 31.9 ± 0.2% to 49.3 ± 0.7% in the G0/G1 phase and decreased it in the S phase from 43.4 ± 0.04 to 40.2 ± 0.3 and in the G2/M phase from 24.7 ± 0.3% to 10.4 ± 0.5%, respectively, by 1.08 and 2.37 folds. Towards SKOV3 cell lines, the population of cells in the G0/G1 phase was not affected for **2f** (31.4 ± 1.2 vs. 30.8 ± 1.5) and increased for **2g** (31.4 ± 1.2 vs. 44.5 ± 0.6). Moreover, in the S phase, it was increased for **2f** (43.1 ± 0.6vs. 59.5 ± 0.2) and not affected for **2g** (43.1 ± 0.6 vs. 44.8 ± 1.6), however, it decreased in the G2/M phase for both **2f** and **2g** by a factor of 2.83 and 2.44, respectively.

Thus, it can be concluded that compound **2f** arrested the cell cycle in the S phase in MCF-7 and SKOV3 cells, and in the G2/M phase in the A549 cell, however, **2g** induced the G0/G1 phase cell cycle arrest and inhibited the progression of the three cancer cells (Table 3).

### 2.3. Computer-Aided Drug Design Studies

#### 2.3.1. Molecular Docking Study

To achieve a better understanding of the expected binding interactions of the target compounds that showed promising IC_50_ values against the three cancer cell lines at the molecular level, analogues **2f** and **2g** were chosen to be docked into the active site of the 3D crystal structure of EGFR (PDB ID: 1M17) (Figure 5). The EGFR is well-known as a receptor that binds to EGFR outside of the cell membrane and is activated, resulting in phosphorylation of the receptor. Phosphorylated EGFR regulates cell survival, proliferation, and metabolism [25,26,27]. Dysregulation of EGFR causing over-expression or constitutive activity leads to cancer in a variety of organs, including the breast, pancreas, lung, and colon. Therefore, EGFR is a much-validated target for oncology drug discovery [28].

MD analyses were carried out using the Glide software to better understand the interactions and affinity of promising compounds **2f** and **2g** with the human EGFR protein. The results revealed that the significant docking scores of 2f and 2g are −5.123 and −6.205 kcal/mol compared to the standard drug Afatinib (−7.886 kcal/mol). The binding interaction showed that compound **2f** formed one hydrogen bond with kinase N-lobes residue Glu738 (ligand OH^….^NH Glu738) at a 1.75 Å bond length, while the central cyclohexane-imidazo ([1,5-b]isoxazol) scaffold formed π-π interaction with Val702, Thr701, Phe699, Gly697, Ser696, Cys773, Gly772, Phe171, Met769, and Leu768 in the N-lobes of the EGFR kinase domain (Figure 6A). A salt bridge between two highly conserved side chains (Lys721-Glu738) that interacts with the α- and β-phosphates when ATP is present is one of the essential traits that mark the N-lobes of active forms of kinases [29]. On the other hand, the methyl 2-hydroxy-5-methylbenzoate group of 2g showed hydrogen bonding with Asp831 (ligand OH^….^NH Asp831) with a 1.96 A° bond length, which is the important element of the catalytic machinery bordering the cleft and the A-loop (Asp831-Val852) as well as the C-lobe DFG motif (Asp831-Gly833) [29]. The binding interaction of compounds 2f and 2g revealed that they are profoundly positioned in the catalytic site of the N-lobe and the A-loop C-lobe DFG motif active site, respectively, suggesting their ability to inhibit human EGFR.

#### 2.3.2. Molecular Dynamic Simulations

MD simulations were used to further investigate the interaction between the compounds **2f** and **2g** and their highest docking score poses [30,31,32]. The RMSD of the backbone protein Cα atomic location was determined to study conformational fluctuations and to test the stability of the Apo EGFR, **2f**: EGFR, and **2g**: EGFR complexes. Based on Figure 6A, the protein Cα RMSD of Apo EGFR protein was increased to 2.89 Å at 27.8 ns. Then, till 33 ns with the sharp increscent of 3.08 Å. After that, suddenly it decreased to 2.63 Å from 31.7 to 33.3 ns, respectively. Finally, the RMSD was raised with a maximum value of 4.55 Å, followed by a minimal fluctuation. After binding the EGFR to compounds **2f** and **2g**, the drifts dramatically decreased. There was no significant alteration in protein RMSD values when associated with compounds **2f** and **2g**. The average RMSD values are: (a) **2f**: EGFR complex = 2.868 ± 0.331Å, (b) **2g**: EGFR complex = 2.356 ± 0.342 Å. The extent of the fluctuation and the difference in average RMSD values suggested that the simulation maintained stable trajectories.

RMSF with respect to the average conformation from MD simulation is used to describe variations in residue flexibility [33]. As depicted in Figure 6B, the RMSF of the protein Cα atoms of each residue in the EGFR: compound **2f** and/or compound **2g** complex, and also in free EGFR, is calculated to exhibit the flexibility of the backbone structure. The mean RMSF value of free EGFR is 1.579 Å. The values for **2f**: EGFR and **2g**: EGFR complex are 3.143 and 3.459 Å, respectively. The least-fluctuating residues are those found in the side chains of the catalytic machinery (Lys721-Glu738), the C-lobe DFG motif (Asp831-Gly833), and the A-loop (Asp831-Val852), which indicates the rigidity of the residues. Higher fluctuation is shown in the C and N terminals of the EGFR protein, where residues are not implicated in the binding of compounds **2f** and **2g**.

H-bond formation minimizes the system’s energy. The H-bond count of the complex fluctuates between 0–3 and 0–4 with an average value of 1.30 and 1.07 for **2f**: EGFR and **2g**: EGFR complexes, respectively. Compound **2f** formed slightly more hydrogen bonds with the EGFR protein than **2g**, which indicates there is slightly more binding activity occurring in the compound **2f**. The trajectory tracing H-bond number is illustrated in Figure 6C. To comprehend the degree of compaction in the structure of EGFR in the presence and absence of an inhibitor, the radius of gyration (RGyr) of EGFR complexes was also calculated. The RMSD of a group of atoms from their shared center of mass is given as the RGyr [34]. The Rg parameter may thus be used to investigate the overall dimensions and changes to the macromolecular structure throughout the MD simulation. Figure 6D shows the computed and plotted RGyr values for each frame of free protein and both complexes versus the simulation time. The RGyr value varied from 3.681 and 4.017 Å, and 3.940 and 4.325 Å for the **2f**: EGFR and **2g**: EGFR complex, respectively, while Apo EGFR had 3.872 and 4.351 Å RGyr (Figure 6D). The RGyr parameter of the EGFR system bound with **2g** was fluctuated with 4.1 Å till 51ns then steadily decreased from approximately 4.107 Å and finally equilibrated at approximately 4.06 Å. With the exception of a few oscillations at approximately 95 ns, the RGyr in the instance of the **2f**: EGFR complex stayed constant throughout the simulation. As a result, the trend of the RGyr plot versus simulation time and its low fluctuation surely explain the residual backbone and how folding of the EGFR protein remained stable after interaction with compounds **2f** and **2g**.

#### 2.3.3. In Silico Evaluation of Physicochemical and ADME Properties [35,36,37,38,39,40,41]

An in silico study (Table 4) of the most biologically active synthesized compounds **2f** and **2g** has been assessed by Swiss ADME server. Lipinski’s rule states that both investigated molecules satisfied this rule and thereby should be taken orally. The blood–brain permeation barrier (BBB permeant) revealed that **2f** crosses the BBB. However, **2g** could not cross it, implies that the later acted by decreasing the central nervous system’s (CNS) side effects. Compounds **2f** and **2g** with consensus log Po/w values of 3.31 and 3.79, respectively, demonstrate good lipophilic character. They also displayed high gastrointestinal absorption and are not P-gp substrate sharing a bioavailability score of 0.55. The interaction of the examined compounds with cytochromes P450 (CYP) isoenzymes which are an important set of enzymes involved in the metabolism of various drugs, essentially included in Phase I biotransformation showed variable behavior. The pink area of the radar chart (Table 4) outlined that both compounds were included in the same area suggesting their good oral bioavailability, as predicted. The BOILED-Egg graph (Table 4) clearly justified the aforementioned results. Our findings revealed that **2f** and **2g** could be good templates for further drug discovery and development.

## 3. Materials and Methods

### 3.1. Chemistry

#### General Synthetic Procedure

Nitrone (1 mmol) and alkene (1 mmol) in 5 mL of toluene were successively placed in a 10 mL quartz cylindrical tube and introduced into a 200 microwave reactor. The irradiation was carried out at 130 °C for 2 h.

(2S,2′R,3a′S,5R)-2′-(4-hydroxy-3-methoxybenzyl)-2-isopropyl-5,5′-dimethyldihydro-2′H-spiro[cyclohexane-1,6′-imidazo [1,5-b]isoxazol]-4′(5′H)-one (**2a**).

Yield 86%; colorless oil; 1H NMR (CDCl_3_, 400 MHz) δ_ppm_ 0.70 (d, 3H, *J* = 6.4 Hz); 0.81 (d, 3H, *J* = 6.8 Hz); 0.83 (d, 3H, *J* = 6.8 Hz); 0.86 (m, 1H); 1.10 (t, 1H, *J* = 12.4 Hz); 1.29 (dd, 1H, *J* = 3.2 and 12 Hz); 1.36 (quin, 1H, *J* = 6.8 Hz); 1.57 (dd, 1H, *J* = 3.2 and 13.2 Hz); 1.68 (m, 2H); 1.76 (m, 1H); 1.93 (m, 1H); 2.18 (ddd, 1H, *J* = 2.8 Hz, 8.8 Hz, 11.6 Hz); 2.66 (m, 1H); 2.67 (s, 3H, NCH_3_); 2.74 (brd, 2H, *J* = 6.4 Hz); 3.77 (ddd, 1H, *J* = 4.8 Hz, 5.6 Hz, 10.4 Hz); 3.82 (s, 3H, OCH_3_); 3.98 (d, 1H, *J* = 8.4 Hz); 5.76 (brs, 1H, OH); 6.67 (m, 2H); 6.78 (d, 1H, *J* = 6 Hz).

13C NMR (CDCl_3_, 100 MHz) δ_ppm_ 18.3; 21.1; 21.2; 24.0; 24.2; 25.9; 29.6; 34.4; 38.4; 38.7; 40.2; 47.9; 55.7; 66.3; 78.7; 99.0; 111.6; 114.2; 121.3; 130.2; 144.1; 146.2; 172.8. Anal. Calcd. for C_23_H_34_N_2_O_4_ (402.54): C, 68.63; H, 8.51; N, 6.96, Found: C, 68.97; H, 8.66; N, 7.08.

(2R,2′R,3a′S,5R)-2′-(2-acetyl-6-hydroxybenzyl)-2,5,5′-trimethyldihydro-2′H-spiro[cyclohexane-1,6′-imidazo [1,5-b]isoxazol]-4′(5′H)-one (**2b**).

Yield 84%; colorless oil; 1H NMR (CDCl_3_, 400 MHz) δ 0.78 (d, 3H, *J* = 6.4 Hz); 0.83 (d, 3H, *J* = 6.4 Hz); 0.85 (d, 3H, *J* = 6.4 Hz); 0.89 (m, 1H); 1.19 (t, 1H, *J* = 12.3 Hz); 1.37 (m, 2H); 1.63 (m, 1H); 1.71 (m, 1H); 1.83 (brd, 1H, *J* = 13.2 Hz); 1.89 (m, 1H); 2.30 (ddd, 1H, *J* = 7.2, 8.8 and 12.0 Hz); 2.53 (s, 3H, CH_3_); 2.69 (m, 1H); 2.71 (s, 3H, NCH_3_); 2.84 (dd, 1H, *J* = 8 and 14.8 Hz); 3.02 (dd, 1H, *J* = 3.2 and 14.8 Hz); 4.03 (d, 1H, *J* = 8.4 Hz); 4.07 (m, 1H), 6.92 (d, 1H, *J* = 8.4 Hz); 7.73 (d, 1H, *J* = 2 Hz); 7.77 (dd, 1H, *J* = 2 and 8.4 Hz); 8.29 (brs, 1H, OH).

13C NMR (CDCl_3_, 100 MHz) δ_ppm_ 18.4; 22.0; 22.1; 24.1; 24.3; 26.1; 26.3; 29.4; 34.2; 34.5; 37.9; 40.4; 48.0; 66.2; 78.0; 90.1; 116.5; 124.2; 129.7; 129.9; 132.1; 160.0; 172.6; 196.9. Anal. Calcd. for C_24_H_34_N_2_O_4_ (414.55): C, 69.54; H, 8.27; N, 6.76, Found: C, 69.74; H, 8.33; N, 6.83.

(2S,2′R,3a′S,5R)-2′-(2-hydroxy-3-methylbenzyl)-2-isopropyl-5,5′-dimethyldihydro-2′H-spiro[cyclohexane-1,6′-imidazo [1,5-b]isoxazol]-4′(5′H)-one (**2c**).

Yield 85%; colorless oil; 1H NMR (CDCl_3_, 400 MHz) δ 0.79 (d, 3H, *J* = 6.4 Hz); 0.84 (d, 3H, *J* = 6.4 Hz); 0.85 (d, 3H, *J* = 6.8 Hz); 0.91 (dd, 1H, *J* = 3.6 and 12.4 Hz); 1.17 (t, 1H, *J* = 12.4 Hz); 1.34 (dd, 1H, *J* = 3.2 and 12 Hz); 1.40 (quin, 1H, *J* = 6.8 Hz); 1.62 (m, 1H); 1.74 (ddd, 1H, *J* = 3.2, 10 and 12.8 Hz); 1.83 (m, 2H); 1.94 (m, 1H); 2.24 (s, 3H, CH_3_); 2.30 (ddd, 1H, *J* = 5.6, 8.8, and 12 Hz); 2.69 (s, 3H, NCH_3_); 2.73 (m, 1H); 2.85 (m, 2H); 3.39 (d, 1H, *J* = 5.2 Hz); 4.02 (brd, 1H, 8 Hz); 4.05 (m, 1H); 6.74 (t, 1H, *J* = 7.6 Hz); 6.86 (d, 1H, 7.4 Hz); 7.01 (d, 1H, *J* = 8.1 Hz); 7.04 (brs, 1H, OH).

13C NMR (CDCl_3_, 100 MHz) δ_ppm_ 16.2; 18.4; 22.0; 24.1; 24.2; 26.0; 29.3; 34.5; 34.8; 38.0; 40.3; 48.0; 53.9; 66.2; 78.9; 89.9; 120.0; 123.9; 125.9; 128.4; 129.8; 153.4; 172.4. Anal. Calcd. for C_23_H_34_N_2_O_3_ (386.54): C, 71.47; H, 8.87; N, 7.25, Found: C, 71.80; H, 8.93; N, 7.37.

(2S,2′R,3a′S,5R)-2′-(3,4-dimethoxybenzyl)-2-isopropyl-5,5′-dimethyldihydro-2′H-spiro[cyclohexane-1,6′-imidazo [1,5-b]isoxazol]-4′(5′H)-one (**2d**).

Yield 83%; colorless oil; 1H NMR (CDCl_3_, 400 MHz) δppm 0.67 (d, 3H, *J* = 6.8 Hz); 0.79 (d, 3H, *J* = 7.2 Hz); 0.81 (d, 3H, *J* = 7.2 Hz); 0.85 (m, 1H); 1.10 (t, 1H, *J* = 12.4 Hz); 1.26 (m, 1H); 1.33 (quin, 1H, *J* = 6.8 Hz); 1.55 (dd, 1H, *J* = 3.2 and 13.6 Hz); 1.65 (m, 2H); 1.74 (m, 1H); 1.90 (m, 1H); 2.15 (ddd, 1H, *J* = 6.4, 8.8 and 12 Hz); 2.62 (m, 1H); 2.64 (s, 3H, NCH_3_); 2.74 (brd, 2H, *J* = 6 Hz); 3.77 (m, 1H); 3.79 (s, 3H, OCH_3_); 3.80 (s, 3H, OCH_3_); 3.95 (d, 1H, *J* = 8.4 Hz); 6.70 (m, 3H).

13C NMR (CDCl_3_, 100 MHz) δppm 18.2; 22.0; 22.1; 24.0; 24.1; 25.8; 29.6; 34.3; 38.3; 38.7; 40.1; 47.8; 55.6; 55.8; 66.2; 78.5; 89.8; 111.0; 112.2; 120.6; 130.9; 147.5; 148.5; 172.6. Anal. Calcd. for C_24_H_36_N_2_O_4_ (416.56): C, 69.20; H, 8.71; N, 6.73, Found: C, 69.49; H, 8.87; N, 6.79.

4-(((1S,2S,2′R,3a′S,5R)-2-Isopropyl-5,5′-dimethyl-4′-oxotetrahydro-2′H-spiro[cyclohexane-1,6′-imidazo [1,5-b]isoxazol]-2′-yl)methyl)-2-methoxyphenyl acetate (**2e**).

Yield 81%; colorless oil; 1H NMR (CDCl_3_, 400 MHz) δppm 0.68 (d, 3H, *J* = 6.4 Hz, CH_3_); 0.78 (d, 3H, *J* = 7.2 Hz); 0.80 (d, 3H, *J* = 6.8 Hz); 0.84 (m, 1H); 1.1 (t, 1H, *J* = 12.4 Hz); 1.26 (m, 1H); 1.34 (m, 1H); 1.55 (dd, 1H, *J* = 2.8, 13.2 Hz); 1.66 (m, 2H); 1.73 (m, 1H); 1.89 (m, 1H); 2.16 (m, 1H); 2.22 (s, 3H, CH_3_); 2.63 (s, 3H, NCH_3_); 2.65 (m, 1H); 2.78 (m, 2H); 3.73 (s, 3H, OCH_3_); 3.81 (m, 1H); 3.95 (d, 1H, *J* = 8.4 Hz); 6.73 (d, 1H, *J* = 8.0 Hz); 6.76 (m, 1H); 6.86 (d, 1H, *J* = 8.0 Hz).

13C NMR (CDCl_3_, 100 MHz) δppm 18.1; 20.4; 21.8; 22.1; 23.9; 24.1; 25.8; 29.5; 34.3; 38.7; 40.1; 47.8; 55.5; 66.0; 78.0; 89.7; 113.0; 120.7; 122.3; 137.2; 138.1; 150.5; 168.7; 172.5. Anal. Calcd. for C_25_H_36_N_2_O_5_ (444.57): C, 67.54; H, 8.16; N, 6.30, Found: C, 67.83; H, 8.35; N, 6.38.

(1S,2S,2′R,3a′S,5R)-2′-(2-Hydroxybenzyl)-2-isopropyl-5,5′-dimethyldihydro-2′H-spiro[cyclohexane-1,6′-imidazo [1,5-b]isoxazol]-4′(5′H)-one (**2f**).

Yield 88%; colorless oil; 1H NMR (CDCl_3_, 400 MHz) δppm δ 0.78 (d, 3H, *J* = 6.4 Hz, CH_3_); 0.82 (d, 3H, *J* = 6.4 Hz); 0.83 (d, 3H, *J* = 6.4 Hz); 0.86 (m, 1H); 1.16 (t, 1H, *J* = 12.4 Hz); 1.34 (m, 2H); 1.59 (dd, 1H, *J* = 2.8 and 13.2 Hz); 1.69 (ddd, 1H, *J* = 2, 10.8 and 12.8 Hz); 1.81 (brd, 2H, *J* = 12.4 Hz); 1.93 (m, 1H); 2.26 (ddd, 1H, *J* = 6.4, 9.2 and 12 Hz); 2.67 (m, 1H); 2.69 (s, 3H, NCH_3_); 2.79 (dd, 1H, *J* = 8.4 and 14.4 Hz); 2.97 (dd, 1H, *J* = 2.8 and 14.4 Hz); 4.01 (d, 2H, *J* = 8.4 Hz); 6.80 (t, 1H, *J* = 7.6 Hz); 6.87 (d, 1H, *J* = 8 Hz); 7.04 (d, 1H, *J* = 8.8 Hz); 7.10 (t, 1H, *J* = 8.4 Hz); 7.45 (brs, 1H, OH).

13C NMR (CDCl_3_, 100 MHz) δppm 18.3; 22.0; 22.1; 24.0; 24.2; 26.0; 29.5; 33.8; 34.4; 37.9; 40.3; 47.9; 66.3; 78.2; 90.0; 116.4; 120.2; 124.2; 128.2; 130.9; 155.0; 172.7. Anal. Calcd. for C_22_H_32_N_2_O_3_ (372.51): C, 70.94; H, 8.66; N, 7.52, Found: C, 71.12; H, 8.71; N, 7.61.

Methyl 2-hydroxy-3-(((1S,2S,2′R,3a′S,5R)-2-isopropyl-5,5′-dimethyl-4′-oxotetrahydro-2′H-spiro[cyclohexane-1,6′-imidazo[1,5-b]isoxazol]-2′-yl)methyl)-5-methylbenzoate (**2g**) [22].

Yield 80%; colorless oil; ^1^H NMR (800 MHz, CDCl_3_) δppm 0.65 (d, 3H, *J* = 6.4 Hz, CH_3_); 0.80 (d, 3H, *J* = 6.4 Hz, CH_3_); 0.82 (d, 3H, *J* = 6.4 Hz, CH_3_); 0.83 (m, 1H); 1.09 (t, 1H, *J* = 11.2 Hz); 1.21 (m, 1H), 1.28 (m, 1H); 1.36 (quin, 1H, *J* = 6.4 Hz); 1.56 (m, 1H); 1.70 (m, 1H); 1.74 (m, 1H); 1.87 (m, 1H); 2.19 (s, 3H, CH_3_); 2.21 (ddd, 1H, *J* = 2.4, 8.0 and 12 Hz); 2.61 (dd, 1H, *J* = 4.8 and 12 Hz); 2.65 (s, 3H, NCH_3_); 2.69 (ddd, 1H, *J* = 5.6, 8.0 and 12 Hz), 3.01 (dd, 1H, *J* = 4.8 and 12 Hz); 3.88 (s, 3H, OCH_3_); 3.96 (m, 2H); 7.17 (d, 1H, *J* = 1.6 Hz); 7.47 (d, 1H, *J* = 1.6 Hz); 10.75 (s, 1H, OH).

^13^C NMR (200 MHz, CDCl_3_) δppm 18.3, 20.2, 21.8, 22.2, 24.0, 24.2, 25.9, 29.4, 32.5, 34.4, 38.5, 40.3, 47.9, 52.1, 66.2, 76.4, 89.7, 111.4, 126.1, 127.5, 127.9, 138.1, 157.4, 170.8, 172.6. Anal. Calcd. for C_25_H_36_N_2_O_5_ (444.57): C, 67.54; H, 8.16; N, 6.30, Found: C, 67.77; H, 8.22; N, 6.39.

### 3.2. Anticancer Activity

#### 3.2.1. Cell Culture

The Roswell Park Memorial Institute medium (RPMI 1640, Gibco, Grand Island, NY, USA) was used to culture the following cancer cell lines including MCF-7, A549, and SKOV3. Cells were grown in FBS (10%) and 100 units mL PS (penicillin/streptomycin) and were supplied to the same medium under standard cell conditions in an incubator (5% of CO_2_, at 37 °C).

#### 3.2.2. Cell Viability Assay

Cell viability was measured according to the previously published work [42].

The EGFR activity was determined based on the method reported by E.A.A. El-Meguid et al. [43].

#### 3.2.3. Cell Cycle Analysis

The effect of selected compounds on cell cycle phase distribution was determined by flow cytometry using the same protocol described elsewhere [44]. Briefly, MCF-7, A549, and SKOV3 followed the same protocol displayed by Elbehairi et al. [44].

### 3.3. Molecular Docking

To understand the binding interaction of promising compounds **2f** and **2g**, a molecular docking study was performed. The crystal structure of the human EGFR bound with erlotinib inhibitor was used in the docking study (PDB ID: 1M17). The study was performed using Schrödinger Glide. The ligand, protein, grid file, and docking approach were all carried out in accordance with our earlier study [45,46,47].

### 3.4. Molecular Dynamic Simulation

MD studies were performed for the docking poses of compounds **2f** and **2g** with the lowest negative scores as described [48,49]. More information on the MD study can be found in previous studies [50,51,52]. The simulation was conducted for 100 ns, generating trajectories of 10,000 frames each for the investigation of protein–ligand interaction dynamics [53,54,55].

### 3.5. ADME Study

SwissADME can be accessed on 3 January 2023 at: http://www.swissadme.ch [56,57].

### 3.6. Acute Toxicity Study

A preliminary acute oral toxicity study of the synthesized compounds was per-formed based on acute toxic class method 423 Guideline, where the toxicity of the most active synthesized compounds (**2f** and **2g)** was tested using three female mice. The mice were fasted prior to dosing (food but water should be withheld) for three hours. After, the animal should be weighted and synthesized compounds were administered initially at a dose of 2000 mg/kg b.w and 1% CMC (p. o.) and were observed for 10 days for acute toxicity. The results revealed no signs of toxicity at 1500 mg/kg b.w. in the experimental animals. Therefore, 75 mg/kg b.w. was selected as the dose for the further studies.

## 4. Conclusions

In the current study, a series of enantiopure isoxazolidine derivatives has been designed and the derivatives evaluated for their anticancer activities towards three human cancer cell lines such as MCF-7, A549, and SKOV3. Compounds **2f** and **2g**, the most active in this study, displayed comparable activity to the standard drug, doxorubicin, with good enzymatic inhibitory activity of EGFR compared to that of Afatinib. Additionally, **2f** and **2g**, the most potent derivatives, were tested for their effect on the cell cycle distribution of the three cell lines, and this study discovered their superior apoptotic effects and that **2f** induced cell cycle arrest at the S phase in MCF-7 and SKOV3 cells, and at G2/M phase in the A549 cell, but **2g** arrested the cell cycle at the G0/G1 phase. Moreover, a docking study of **2f** and **2g** with EGFR revealed that both compounds **2f** and **2g** fit well in the catalytic site of the N-lobe and the A-loop C-lobe DFG motif active site, respectively, suggesting their ability to inhibit human EGFR with good dynamic profile. Otherwise, physicochemical and pharmacokinetic parameters confirm their good bioavailability with acceptable values of drug-likeness. Overall, this study helps us to understand the anticancer mechanisms of **2f** and **2g** that may be developed as a promising class of potential anticancer agents. Furthermore, in vivo assays as well as their efficacy and safety in cancer patients either alone or in combination with other anticancer drugs will be developed in several clinical studies with human subjects.

## Data Availability

Data is contained within the article.

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
