# Peer review of "In Vitro and In Silico Evaluation of Antiproliferative Activity of New Isoxazolidine Derivatives Targeting EGFR: Design, Synthesis, Cell Cycle Analysis, and Apoptotic Inducers"

_pharmaceuticals, 2023, doi:10.3390/ph16071025_

Round 1

Reviewer 1 Report

The manuscript contains some good work. The following, however, should be considered:

1. The sentences lines 63-65 should be rewritten and include modern cancer therapy such as CAR-T and ADC. 

2. There is no control of the effect of the new compounds on healthy cells and how toxic they are. 

3. The authors must comment on new compounds in relation to the current drugs in use and why we need new small molecules. 

4. lines 285-288 need to be rewritten. 

5. Again, most of the manuscript is in long sentences and needs to be reconsidered. 

The quality of the English is reasonable, but it needs a complete revision, especially the length of the sentences. Most of the manuscript was written using long sentences, which caused some confusion. 

Author Response

The manuscript contains some good work. The following, however, should be considered:

- Comments

  1. The sentences lines 63-65 should be rewritten and include modern cancer therapy such as CAR-T and ADC. 

Feedback

We have added the following sentence:

Current ineffective treatments against cancer including surgery, immunosuppression chemotherapy and radiotherapy, have many adverse effects and high drug resistance ac-companied with high rate of incidence, mortality and morbidity [7,8]. In contrast, chimeric antigen receptor-T (CAR-T) cells and antibody-drug conjugates (ADCs) remains currently the most popular immunotherapies strategies in oncology, for anticancer drug development [9,10].

- Comment

  1. There is no control of the effect of the new compounds on healthy cells and how toxic they are. 

Feedback

A preliminary acute oral toxicity study of the synthesized compounds was performed based on acute toxic class method 423 Guideline, where the toxicity of the most active synthesized compounds (2f and 2g) was tested using three female mice. The mice were fasted prior to dosing (food but water should be withheld) for three hours. After, the animal should be weighted and synthesized compounds were administered initially at a dose of 2000mg/kg b.w and 1% CMC (p. o.) and were observed for 10 days for acute toxicity. The results revealed that no sign of toxicity at 1500 mg/kg b.w. in the experimental animals, Therefore, 75 mg/k.g. b.w. was considered as the dose for the further studies.

- Comment

  1. The authors must comment on new compounds in relation to the current drugs in use and why we need new small molecules. 

Feedback

Small molecules are known as landmark drugs in the history of medicine and have played an important role in medical advancements. Also, small molecules have been the right method to optimize drug potency and selectivity and to improve overall molecular properties. Likewise, small molecules represent the suitable candidate as diagnostics, as enablers in breakthrough new technologies, and as scientific tools for chemical biology, such as cell and gene therapy. [REF1].

[REF1] Hartmut Beck, Michael Härter, Bastian Haß, Carsten Schmeck, Lars Baerfacker. Small molecules and their impact in drug discovery: A perspective on the occasion of the 125th anniversary of the Bayer Chemical Research Laboratory. Drug Discovery Today, 2022, 27, 1560-1574. https://doi.org/10.1016/j.drudis.2022.02.015.

- Comment

  1. lines 285-288 need to be rewritten. 

Feedback

Done

- Comment

  1. Again, most of the manuscript is in long sentences and needs to be reconsidered. 

Feedback

Done

- Comments on the Quality of English Language

The quality of the English is reasonable, but it needs a complete revision, especially the length of the sentences. Most of the manuscript was written using long sentences, which caused some confusion.

Feedback:

Done

Reviewer 2 Report

All compounds numbers should be in Bold throughout the text

Line 25 Change to: ...a hydroxyl group...

The sentence in Lines 24-30 is far too long and should be split into 2 or 3 smaller sentences. Also in this sentence, it should be explicit that only compound 2g has an ester group at C-3 of the phenyl ring. 

Lines 44-45 Change to: ...division. It is expected to be the second leading cause of death after cardiovascular disease, with currently about 8 million deaths per year and... 

Line 53 Change to: ...been ranked...

Line 58 Change cancer to: ...cancers.

Line 63 Change ineffectiveness to: ...ineffective...

Line 66 Change agent to: ... agents...

Line 67 Change less to: ... fewer...

Figure 1. Not all compounds depicted are pyridinyl isoxazolidine derivatives. Better give the full or generic chemical name of each individual compound.

Line 78 Change to: ... designed and synthesized...

Line 89 Change Generals to: General Synthetic Procedure 

Line 91-2 Change to: ...10 mL quartz cylindrical tube and introduced into a 200 microwave reactor. The irradiation was carried out...

Give the yield of each compound synthesized in the Material and Methods Section as well as the physical state of each compound i.e. solid or oil. If solid record melting points. 

Change all cases of NMR 1H (CDl3, ... to: 1H NMR (CDl3, ...

Change all cases of NMR 13C (CDl3, ... to: 13C NMR (CDl3, ...

Line 181 Change to: ...the previously published work...

Line 191 Change ...following... to:... followed...

Line 204 Change ...as follows... to: ... as described...

Line 208 Leave space 100 ns

Results and Discussion  Elaborate on the mechanism explaining how a single enantiomer is formed in the reaction of the nitrone 1  the aryl akene.

Line 235 leave space H methyl and H isopropyl (also fix font size)

Line 242  Delete was Change to: ...used as an anticancer...

Lines 240-248 insert space between the value and μM

The sentence in Lines 253-258 is too long. Is difficult to follow and should be split into 2 or 3 smaller sentences.

Line 257 Give the full name of PNA

Line 272 Change to: ... which are potent...

Line 276  Change to: ... 0.484±0.01 μM which was ... delete are

Line 285 is not complete.

Lines 285-288 Change to: In order to elucidate whether cell death was induced by our selected derivatives the effect of compounds 2f and 2g on cellular apoptosis in the three cancer cell lines was evaluated at their IC50 values for 48 h, using the Annexin V and PI double staining assay by flow cytometric analysis, with doxorubicin as a positive control (Figure 3).

Line 298 Change ... reduce to: ... reduced...

 Line 302 Change increase... to: increased...

Line 304 Change ...than... to: ... as...

Line 305 Change ... to induced... to ... induce...

Figures 3 & 4 should be presented in landscape format at a bigger size. As presented is illegible and the font size in the legend is also too small.

Page 11 compound numbers should be in Bold

Page 13 Line 36 Change ...was chosen...to: ... were chosen...

Page 13 Line 49 ...are... not bold

Page 14 Line 73 Then, till33 ns 33 ns ????? Correct this

Page 14 Line 82 delete ...which...

Page 16 Figure 6 same fixing needed as for Figures 3 & 4 

In The Supporting Information Section,  each single NMR spectrum should be in Landscape format on a full page. This will depict the spectra in bigger sizes to make the values readable. It would also be nice to insert the chemical structure of each compound on the top left-hand side of each NMR spectrum. 

The quality of English is acceptable after the corrections I pointed out and final proofreading by a chemist not listed as an author.

Author Response

Reviewer 2

- Comment

All compounds numbers should be in Bold throughout the text

Feedback:

Done

- Comment

Line 25 Change to: ...a hydroxyl group...

Feedback:

Was changed

- Comment

The sentence in Lines 24-30 is far too long and should be split into 2 or 3 smaller sentences. Also in this sentence, it should be explicit that only compound 2g has an ester group at C-3 of the phenyl ring.

Feedback:

Was changed

- Comment  

Lines 44-45 Change to: ...division. It is expected to be the second leading cause of death after cardiovascular disease, with currently about 8 million deaths per year and... 

Feedback:

Was changed

- Comment

Line 53 Change to: ...been ranked...

Feedback:

Was changed

- Comment

Line 58 Change cancer to: ...cancers.

Feedback:

Was changed

- Comment

Line 63 Change ineffectiveness to: ...ineffective...

Feedback:

Was changed

- Comment

Line 66 Change agent to: ... agents...

Feedback:

Was changed

- Comment

Line 67 Change less to: ... fewer...

Feedback:

Was changed

- Comment

Figure 1. Not all compounds depicted are pyridinyl isoxazolidine derivatives. Better give the full or generic chemical name of each individual compound.

Feedback:

Done

- Comment

Line 78 Change to: ... designed and synthesized...

Feedback:

Was changed

- Comment

Line 89 Change Generals to: General Synthetic Procedure 

Feedback:

Was changed

- Comment

Line 91-2 Change to: ...10 mL quartz cylindrical tube and introduced into a 200 microwave reactor. The irradiation was carried out...

Feedback:

Was changed

- Comment

Give the yield of each compound synthesized in the Material and Methods Section as well as the physical state of each compound i.e. solid or oil. If solid record melting points. 

Feedback:

Done

- Comment

Change all cases of NMR 1H (CDl3, ... to: 1H NMR (CDl3, ...

Feedback:

Was changed

- Comment

Change all cases of NMR 13C (CDl3, ... to: 13C NMR (CDl3, ...

Feedback:

Was changed

- Comment

Line 181 Change to: ...the previously published work...

Feedback:

Was changed

- Comment

Line 191 Change ...following... to:... followed...

Feedback:

Was changed

- Comment

Line 204 Change ...as follows... to: ... as described...

Feedback:

Was changed

- Comment

Line 208 Leave space 100 ns

Feedback:

Was changed

- Comment

Results and Discussion:  Elaborate on the mechanism explaining how a single enantiomer is formed in the reaction of the nitrone the aryl akene.

Feedback:

Done

- Comment

Line 235 leave space H methyl and H isopropyl (also fix font size)

Feedback:

Done

- Comment

Line 242  Delete was Change to: ...used as an anticancer...

Feedback:

Was changed

- Comment

Lines 240-248 insert space between the value and μM

Feedback:

Done

- Comment

The sentence in Lines 253-258 is too long. Is difficult to follow and should be split into 2 or 3 smaller sentences.

Feedback:

Done

- Comment

Line 257 Give the full name of PNA

Feedback:

Done

- Comment

Line 272 Change to: ... which are potent...

Feedback:

Was changed

- Comment

Line 276  Change to: ... 0.484±0.01 μM which was ... delete are

Feedback:

Was changed

- Comment

Line 285 is not complete.

Lines 285-288 Change to: In order to elucidate whether cell death was induced by our selected derivatives the effect of compounds 2f and 2g on cellular apoptosis in the three cancer cell lines was evaluated at their IC50 values for 48 h, using the Annexin V and PI double staining assay by flow cytometric analysis, with doxorubicin as a positive control (Figure 3).

Feedback:

Done

- Comment

Line 298 Change ... reduce to: ... reduced...

Feedback:

Was changed

- Comment

 Line 302 Change increase... to: increased...

Feedback:

Was changed

- Comment

Line 304 Change ...than... to: ... as...

Feedback:

Was changed

- Comment

Line 305 Change ... to induced... to ... induce...

Feedback:

Was changed

- Comment

Figures 3 & 4 should be presented in landscape format at a bigger size. As presented is illegible and the font size in the legend is also too small.

Feedback:

Done

- Comment

Page 11 compound numbers should be in Bold

Feedback:

Done

- Comment

Page 13 Line 36 Change ...was chosen...to: ... were chosen...

Feedback:

Was changed

- Comment

Page 13 Line 49 ...are... not bold

Feedback:

Done

- Comment

Page 14 Line 73 Then, till33 ns 33 ns ????? Correct this

Feedback:

Done

- Comment

Page 14 Line 82 delete ...which...

Feedback:

Done

- Comment

Page 16 Figure 6 same fixing needed as for Figures 3 & 4 

Feedback:

Done

- Comment

In The Supporting Information Section,  each single NMR spectrum should be in Landscape format on a full page. This will depict the spectra in bigger sizes to make the values readable. It would also be nice to insert the chemical structure of each compound on the top left-hand side of each NMR spectrum. 

Feedback:

Done

Reviewer 3 Report

The work with title “In vitro and in silico evaluation of antiproliferative activity of new isoxazolidine derivatives targeting EGFR: Design, synthesis, cell cycle analysis and apoptotic inducers” is a valuable and suitable contribution to be published in Pharmaceuticals journal after justifying some points.

This work involving the synthesis and evaluation of novel enantiopure isoxazolidine derivatives for their anticancer activities. The researchers synthesized a series of novel isoxazolidine derivatives and tested their anticancer activities using the MTT assay. The compounds were evaluated against three different human cancer cell lines: MCF-7, A-549, and SKOV3. Two specific compounds, 2f and 2g, showed significant inhibition of cell growth. Compound 2f, with a hydroxyl group attached to the C-2 position of the phenyl ring, and compound 2g, with an ester group at the C-3 position and a small electron-donating group (methyl) at the C-5 position of the phenyl ring, demonstrated potent anticancer effects. The inhibitory activity was dose-dependent, and the IC50 values for these compounds were provided for each cell line.

Overall, this work highlighting the promising anticancer activities, EGFR inhibitory effects, cell cycle arrest, apoptotic effects, binding affinity, pharmacokinetic properties, and conformational stability of the synthesized isoxazolidine derivatives. which has strong impact to be readable and attractive for the other researchers.  

·       The similarity rate should be reduced as much as possible

·       about 21 references were belongs to the main author which should be reduced as possible regarding the self-citation issue.

·       Some typo and language errors were detected showed be corrected accordingly

·       Line 21 it is recommended to use evaluated instead of tested.

·       You can add to the abstract a sentence regarding the characterization methods for these new compounds like NMR and HRMS.  

·       Line 27 +28 and in the whole MS you should write IC50 as subscript for 50

·       Line 31 add “respectively” after the values, and add spaces before the units.

·       Remove the IC50 from the keywords because it is presents in almost all other works  

·       It is recommended to add the last updates of cancer statistics and cite some recent work, (Biomolecules 2022, 12, 1843) which may improve the introduction section well.

·      Regarding the isoxazole derivatives with various biological activities you can add the following recent works which can reduce the self-citation percentage: Synthesis of novel isoxazole–carboxamide derivatives as promising agents for melanoma and targeted nano-emulgel conjugate for improved cellular permeability. BMC Chemistry 16, 47 (2022), 3 Biotech (2022) 12:342 Molecular docking studies and biological evaluation of isoxazole‑carboxamide derivatives as COX inhibitors and antimicrobial agents.

·       Line 92 correct “130C for 2 hours” as “130 Ö¯C for 2 hours.

·       Line 97+108 and in the whole MS the H NMR should be written as 1H-NMR, for 1 as superscript. as well as for C NMR

·       The “ J” coupling constant should be in italic

·   In some NMR spectrums there is some un recorded signals and does not belong to your compounds why?

·     Line 97+108+114..etc,  you have to add “:“ after NMR 1H (CDCl3, 400 MHz) δppm  

·       Line 216 and 218 edit to 1a-1g and 2a-2g

·       Line 239 edit in vitro to italic

·       Lines 240-252 re-write this paragraph again it is confusing.

·       Better resolution of Figure 3 is recommended

·       In figure 5 what is the meaning of the blue arrow, it is a hydrogen bond ?? if yes, add that to the caption of this figure, as well as add the other binding interactions to the 2D.

·       Regarding the Pharmacokinetics and BOILED-Egg model discuss that with a recent similar works Design, synthesis, molecular docking and biological evaluation of new carbazole derivatives as anticancer, and antioxidant agents. BMC Chemistry 17, 60 (2023), which could improve the discussion section.

·       Line 145 of Conclusions remove the “and “ namely MCF-7 and A549 and SKOV3 where

·       Improve the conclusion section with possible future works according in vivo or promising more similar series.

·       In overall, the work appears to have scientific merit and contributes to the development of new anticancer agents.

Best wishes

Author Response

Reviewer 3

- Comment  

  • The similarity rate should be reduced as much as possible

Feedback:

Done

     - Comment

  • about 21 references were belongs to the main author which should be reduced as possible regarding the self-citation issue.

Feedback:

Done

     - Comment

  • Some typo and language errors were detected showed be corrected accordingly

Feedback:

Done

    - Comment

         Line 21 it is recommended to use evaluated instead of tested.

Feedback:

Was changed

     - Comment

  • You can add to the abstract a sentence regarding the characterization methods for these new compounds like NMR and HRMS. 

Feedback:

Done

      - Comment

  • Line 27 +28 and in the whole MS you should write IC50 as subscript for 50

Feedback:

Done

      -Comment

  • Line 31 add “respectively” after the values, and add spaces before the units.

Feedback:

Done

      - Comment

  • Remove the IC50 from the keywords because it is presents in almost all other works

Feedback:

Done

 - Comment

  • It is recommended to add the last updates of cancer statistics and cite some recent work, (Biomolecules 2022, 12, 1843) which may improve the introduction section well.

Feedback

Done

- Comment

  • Regarding the isoxazole derivatives with various biological activities you can add the following recent works which can reduce the self-citation percentage: Synthesis of novel isoxazole–carboxamide derivatives as promising agents for melanoma and targeted nano-emulgel conjugate for improved cellular permeability. BMC Chemistry 16, 47 (2022), 3 Biotech (2022) 12:342 Molecular docking studies and biological evaluation of isoxazole‑carboxamide derivatives as COX inhibitors and antimicrobial agents.

Feedback

Done

    - Comment

  • Line 92 correct “130Cfor 2 hours” as “130 Ö¯C for 2 hours”.

Feedback:

Done

    - Comment

  • Line 97+108 and in the whole MS the H NMR should be written as 1H-NMR, for 1 as superscript. as well as for C NMR

Feedback:

Done

     - Comment

  • The “ J” coupling constant should be in italic

Feedback:

Done

     -  Comment

  • In some NMR spectrums there is some un recorded signals and does not belong to your compounds why?

Feedback:

       Some NMR spectra contain traces of solvent namely compound 2b

     - Comment

  • Line 97+108+114..etc,  you have to add “:“ after NMR 1H (CDCl3, 400 MHz) δppm

Feedback:

       Done

     - Comment

  • Line 216 and 218 edit to 1a-1gand 2a-2g

Feedback:

Was changed

     - Comment

  • Line 239 edit in vitro to italic

Feedback:

Was changed

- Comments

  • Lines 240-252 re-write this paragraph again it is confusing.

Feedback:

Done

- Comments

  • Better resolution of Figure 3 is recommended

Feedback:

Done

- Comments

  • In figure 5 what is the meaning of the blue arrow, it is a hydrogen bond ?? if yes, add that to the caption of this figure, as well as add the other binding interactions to the 2D.

Feedback:

Done

- Comment

  • Regarding the Pharmacokinetics and BOILED-Egg model discuss that with a recent similar works Design, synthesis, molecular docking and biological evaluation of new carbazole derivatives as anticancer, and antioxidant agents. BMC Chemistry 17, 60 (2023), which could improve the discussion section.

Feedback:

Done

- Comment

  • Line 145 of Conclusions remove the “and “ namely MCF-7 and A549 and SKOV3 where

Feedback:

Done

- Comment

Improve the conclusion section with possible future works according in vivo or promising more similar series.

Feedback:

Done

Round 2

Reviewer 1 Report

The authors have responded to some questions. The authors, however, responded to my point 2 regarding the effect of the compounds on healthy cells in their response, but I could not see any mention of the effect of these compounds in the manuscript. The authors should have included healthy cells in their apoptosis study. These compounds could be less beneficial if they showed toxicity against healthy cells. 

The authors responded to point 3 in general terms. Small molecules are effective drugs, but they are much less specific. The authors must comment on why their compounds are useful in relation to the known drugs.      

Minor  mistakes to be corrected 

Author Response

Point by point response to reviewer’s Comments

Manuscript ID: pharmaceuticals-2470430

Title: In vitro and in silico evaluation of antiproliferative activity of new
isoxazolidine derivatives targeting EGFR: Design, synthesis, cell cycle
analysis and apoptotic inducers

We wish to express our appreciation to the Editor and Reviewers for their insightful precious time and invaluable comments. We have carefully addressed all the comments. The corresponding changes and refinements made in the revised paper are summarized in our response below and were cited in the manuscript in BLUE color.

Point-by-point responses to the comments are listed below.

Sincerely

Prof. Kaiss Aouadi

Comment

The authors have responded to some questions. The authors, however, responded to my point 2 regarding the effect of the compounds on healthy cells in their response, but I could not see any mention of the effect of these compounds in the manuscript. The authors should have included healthy cells in their apoptosis study. These compounds could be less beneficial if they showed toxicity against healthy cells.

Feedback

We have added in the text the following sentence:

A preliminary acute oral toxicity study of the synthesized compounds was performed based on acute toxic class method 423 Guideline, where the toxicity of the most active synthesized compounds (2f and 2g) was tested using three female mice. The mice were fasted prior to dosing (food but water should be withheld) for three hours. After, the animal should be weighted and synthesized compounds were administered initially at a dose of 2000mg/kg b.w and 1% CMC (p. o.) and were observed for 10 days for acute toxicity. The results revealed that no sign of toxicity at 1500 mg/kg b.w. in the experimental animals, Therefore, 75 mg/k.g. b.w. was considered as the dose for the further studies.

Comment

The authors responded to point 3 in general terms. Small molecules are effective drugs, but they are much less specific. The authors must comment on why their compounds are useful in relation to the known drugs.

Feedback

Research on molecularly targeted cancer drug discovery over the last few decades has resulted in a number of small molecule drugs being successfully introduced in the clinic for cancer treatment. Among them and similar to our work, isoxazolidine derivatives have been approved for their promising activity against cancer cell lines such as murine leukemia L1210, human lymphocyte CEM and human cervix carcinoma HeLa cells [REF].

REF. Kokosza, K., Andrei, G., Schols, D., Snoeck, R., Piotrowska, D.G., Design, antiviral and cytostatic properties of isoxazolidine-containing amonafide analogues, Bioorganic & Medicinal Chemistry (2015), doi: http://dx.doi.org/10.1016/j.bmc.2015.04.079

Comment

Minor mistakes to be corrected

Feedback

The manuscript has been revised and errors have been corrected

Round 3

Reviewer 1 Report

Ready to be published; thank you. 

Author Response

We thank the referee for the careful and insightful review of our manuscript.